# Impact of *CARD14 rs34367357* Mutation, Nutrition Status, and Seasonality on the Response to Biologic Therapy in Psoriasis—A Retrospective Observational Single-Center Study

**DOI:** 10.3390/jcm14186458

**Published:** 2025-09-13

**Authors:** Michał Niedźwiedź, Agnieszka Czerwińska, Janusz Krzyścin, Izabela Dróżdż, Sebastian Skoczylas, Joanna Narbutt, Aleksandra Lesiak

**Affiliations:** 1Department of Dermatology, Paediatric Dermatology and Oncology, Medical University of Lodz, 90-419 Lodz, Poland; joanna.narbutt@umed.lodz.pl (J.N.); aleksandra.lesiak@umed.lodz.pl (A.L.); 2International Doctoral School, Medical University of Lodz, 90-419 Lodz, Poland; 3Institute of Geophysics, Polish Academy of Sciences, 01-452 Warsaw, Poland; aczerwinska@igf.edu.pl (A.C.); jkrzys@igf.edu.pl (J.K.); 4Department of Clinical Genetics, Medical University of Lodz, 92-215 Lodz, Poland; izabela.drozdz@umed.lodz.pl (I.D.); sebastian.skoczylas@umed.lodz.pl (S.S.); 5Laboratory of Autoinflammatory, Genetic and Rare Skin Disorders, Medical University of Lodz, 92-215 Lodz, Poland

**Keywords:** psoriasis, biological therapy, *CARD14 rs34367357*, seasonality, personalized treatment, nutrition status, clinical outcomes

## Abstract

**Background/Objectives:** Psoriasis is an immune-mediated disease influenced by genetic predisposition, environmental triggers, and metabolic comorbidities. Biologic therapies have markedly improved disease control; however, variability in patient response remains insufficiently understood. The aim of the study is to evaluate whether *CARD14* mutations and the season of treatment initiation influence the efficacy of biologic therapy in psoriasis. We also examined the potential interactions between *CARD14* status, seasonality, drug class, and nutrition status on short-term clinical outcomes. **Methods:** This retrospective study included 72 patients receiving biologic therapy within the Polish NHF B.47 program. Clinical endpoints (PASI, BSA, DLQI) were assessed at baseline and after 1, 4, 7, and 10 months. *CARD14* genotyping was performed using Sanger sequencing. Patients were stratified according to mutation status, season of therapy initiation (warm vs. cold), drug class, and BMI category. Statistical analyses included *t*-tests, chi-square, ANOVA, and MANOVA. **Results:** The *CARD14 rs34367357* mutation was associated with earlier disease onset (15.6 vs. 22.7 years, *p* = 0.0134) and higher DLQI baseline (*p* = 0.0265) but did not significantly impact treatment response. Therapy initiated during the warm season (April–September) led to greater PASI improvement (*p* < 0.0001). Obesity was associated with reduced response (*p* = 0.02385). Drug class and interaction effects were not statistically significant. **Conclusions:** Our findings suggest that seasonal timing, nutritional status, and genetic background may modulate the efficacy of biologic therapies in psoriasis. Although not statistically conclusive, the potential interaction between *CARD14 rs34367357* and seasonality warrants further investigation.

## 1. Introduction

Psoriasis is a chronic, immune-mediated skin disease that affects approximately 2–3% of the global population, with significant geographical and ethnic variation in prevalence [1]. Clinically, it is characterized by erythematous, scaly plaques, most commonly on the extensor surfaces, scalp, and trunk. Although psoriasis has long been considered a dermatological disorder, it is now recognized as a systemic inflammatory condition frequently associated with comorbidities such as psoriatic arthritis, obesity, metabolic syndrome, cardiovascular disease, and depression [2,3,4]. The disease imposes a substantial burden on patients’ physical, psychological, and social well-being, which in turn translates into reduced quality of life and increased healthcare utilization [5].

In recent years, the management of moderate-to-severe psoriasis has been revolutionized by the introduction of biologic therapies targeting specific cytokines implicated in disease pathogenesis, including tumor necrosis factor alpha (TNF-α), interleukin-12/23 (IL-12/23), IL-17, and IL-23 [6,7]. These agents have demonstrated remarkable efficacy in reducing disease activity and improving patient-reported outcomes, with response rates frequently surpassing those of traditional systemic agents such as methotrexate or cyclosporine [1]. However, not all patients respond equally to biologic treatment. In clinical practice, a substantial proportion of patients fail to achieve complete or sustained remission, and primary or secondary non-response remains a significant challenge [8,9].

Understanding the sources of variability in treatment response is thus of considerable clinical and scientific interest. Emerging evidence suggests that both intrinsic and extrinsic factors may modulate the efficacy of biologic therapies. Hormonal and sex-related differences are increasingly recognized as important modulators of psoriasis course and treatment efficacy [10,11]. Estrogen and androgen pathways influence keratinocyte proliferation and immune responses, contributing to sex-specific patterns of disease activity [10,12,13]. Recent evidence confirms that sex may impact treatment outcomes in patients receiving biologic therapy [10,11,12,13,14]. Therefore, consideration of both genetic and sex-related factors is critical for understanding variability in treatment response.

Genetic predisposition is a key determinant in psoriasis pathogenesis, and several loci have been identified as risk factors. Among them, *CARD14* encodes a keratinocyte protein involved in NF-κB signaling, and variants within this gene have been associated with psoriasis susceptibility, particularly in patients with early-onset and familial disease [15,16]. The *rs34367357* polymorphism has been described as a functionally relevant variant, potentially contributing to aberrant epidermal signaling and immune activation [16]. Given these associations, we hypothesized that *CARD14 rs34367357* might influence not only the onset of psoriasis but also the response to biologic therapies. Among the intrinsic factors, genetic predisposition has been shown to influence not only susceptibility to psoriasis but also its severity and treatment outcomes. In particular, variants in genes involved in the IL-23/Th17 axis, antigen presentation, and keratinocyte signaling have attracted attention as potential biomarkers of response [17]. One such gene is *CARD14* (caspase recruitment domain family member 14), which encodes a protein involved in NF-κB activation in keratinocytes. Mutations in *CARD14* have been associated with familial psoriasis, early-onset disease, and pustular variants [15,18].

In parallel, environmental factors—including seasonality—have been proposed as important modulators of disease activity and treatment response [19,20]. Psoriasis is known to exhibit seasonal fluctuations, with symptoms commonly improving during the summer months and worsening in winter. These changes have been attributed to increased sunlight exposure, higher vitamin D levels, and reduced environmental triggers such as infections or dry air [20]. Several studies have noted higher rates of disease exacerbation and hospital admissions in colder seasons [20,21,22]. However, the potential influence of seasonality on treatment initiation and subsequent biologic response remains underexplored [23].

Furthermore, obesity has emerged as a critical factor in psoriasis pathogenesis and management. Adipose tissue acts as a reservoir for proinflammatory cytokines, and elevated body mass index (BMI) has been associated with increased disease severity and diminished response to biologic therapy [3,24]. Despite its clear clinical relevance, the interplay between obesity, genetics, environmental exposure, and treatment timing has received limited systematic attention.

In this study, we aim to investigate the relationships between *CARD14* mutation status, seasonal timing of treatment initiation, nutrition status, and clinical response to biologic therapy in patients with moderate-to-severe plaque psoriasis. We assessed whether these factors independently or interactively influenced initial treatment outcomes measured by Psoriasis Area and Severity Index (PASI), Body Surface Area (BSA), and Dermatology Life Quality Index (DLQI) within the first 10 months of therapy. By integrating genetic, environmental, and clinical data, we sought to identify potential biomarkers and contextual modifiers that could inform a more personalized approach to psoriasis treatment.

## 2. Aims

The primary aim of this study is to assess whether the presence of the *CARD14* mutations and the season of biologic treatment initiation affect clinical response in patients with moderate-to-severe plaque psoriasis. Secondary aims include evaluating the baseline differences in disease severity and quality of life between *CARD14* mutations carriers and wild-type individuals, assessing whether seasonality, obesity, or drug class modifies short-term treatment outcomes, and identifying potential interactions among genetic, environmental, and therapeutic variables using multivariate analysis.

## 3. Material and Methods

### 3.1. Study Design and Population

This was a single-center, retrospective study including 72 adult patients with moderate-to-severe plaque psoriasis who received biologic therapy under the B.47 therapeutic program financed by the Polish National Health Fund. All patients fulfilled the eligibility criteria for biologic treatment, including PASI ≥ 10 and BSA ≥ 10%, and failure or intolerance to conventional systemic therapy. Data were collected between March 2021 and March 2022. Genomic DNA was isolated from peripheral blood samples using standard phenol-chloroform extraction. Genotyping of the *CARD14* gene was performed via Sanger sequencing. In accordance with the laboratory’s standard operating procedures, each sample was tested in triplicate to ensure reproducibility and reliability of the results.

### 3.2. Clinical Assessment

Demographic and clinical data were retrieved from medical records. Psoriasis severity was measured using PASI, BSA and DLQI scores at baseline and at months 1, 4, 7, and 10 following treatment initiation. Additional variables included age, sex, BMI, comorbidities, and family history. BMI was categorized according to simplified WHO definitions into normal weight (<25), and overweight and obese (25≥). Seasonality was defined as “warm season”: April to September and “cold season”: October to March. Drug classes included TNF-α inhibitors (adalimumab, etanercept), IL-12/23 (ustekinumab), IL-17 (ixekizumab, and secukinumab), and IL-23 inhibitors (guselkumab, risankizumab, and tildrakizumab). All biologic therapies were administered strictly according to the Summary of Product Characteristics (SMPC) for each agent, within the framework of the NHF B.47 program. No off-label dose adjustments were performed during the observation period.

### 3.3. Statistical Analysis

Continuous variables were presented as means and standard deviations or medians and interquartile ranges, depending on distribution. Categorical variables were expressed as counts and percentages. Group comparisons were made using Student’s *t*-tests or Mann–Whitney U tests, as appropriate. Chi-square tests assessed categorical associations. Longitudinal changes in PASI, BSA, and DLQI were analyzed using repeated-measures ANOVA. MANOVA was used to test multivariate effects and interactions. We performed a complete-case analysis: only participants with full PASI, BSA, and DLQI data across all scheduled timepoints (0, 1, 4, 7, and 10 months) were included. Due to mandatory program visits, missingness was negligible; no imputation or list-wise/case-wise deletion was required. A *p*-value < 0.05 was considered statistically significant. All analyses were conducted using TIBCO Statistica, v. 13.5.0, TIBCO Software Inc., Palo Alto, CA, USA software.

## 4. Results

A total of 72 patients diagnosed with moderate-to-severe psoriasis were included in the analysis (Table 1). Fifteen patients (20.8%) were carriers of the *CARD14 rs34367357* mutation, while 57 (79.2%) were wild-type. Comparative analyses were conducted to evaluate differences in demographics, clinical characteristics, treatment allocation, and short-term treatment response. The distribution of sex and mean BMI did not significantly differ between groups. Therefore, any variations in treatment courses are more likely attributable to genetic or environmental factors rather than differences in nutritional status. Patients with the mutation exhibited a significantly younger age of psoriasis onset (13.9 ± 8.4 vs. 22.4 ± 14.9 years, *p* = 0.0134), while the age at treatment initiation was comparable (*p* = 0.7887). Baseline disease severity as measured by PASI and BSA scores was similar in both groups, but baseline quality of life, reflected by DLQI, was significantly worse among mutation carriers (23.5 ± 4.5 vs. 19.9 ± 4.3, *p* = 0.0265). Regarding biologic treatments, both groups showed comparable distribution across drug classes. TNF-alpha inhibitors were the most frequently used in both cohorts (36 vs. 7 patients), followed by IL-17 and IL-12/23 inhibitors. IL-23 inhibitors were used exclusively in the non-mutant group. Secukinumab and adalimumab were the most used individual drugs across both subgroups.

Sex did not significantly influence treatment outcomes across the analyzed endpoints. Repeated-measures ANOVA showed no significant interaction between sex and time for PASI (*p* = 0.1825), BSA (*p* = 0.9138), or DLQI (*p* = 0.9337). Both male and female patients demonstrated comparable improvement trajectories over the 10-month observation period.

To evaluate the impact of the *CARD14 rs34367357* mutation on treatment efficacy, we compared clinical scores over time between mutation carriers and wild-type individuals. PASI, BSA, and DLQI were assessed at baseline and after 1, 4, 7, and 10 months of therapy.

The dynamics of PASI reduction (Figure 1) showed a consistent downward trend in both groups, with no statistically significant difference between mutation carriers and non-carriers (F (4, 280) = 1.0558, *p* = 0.37875). However, patients with the *CARD14* mutation tended to reach lower PASI scores by month 10, suggesting a slightly better response, though this observation did not reach statistical significance. BSA improvement was significantly more noticeable in the mutation group (F (4, 280) = 2.9352, *p* = 0.02110), particularly from month 4 onward. This indicates a greater and earlier reduction in the extent of skin involvement among mutation carriers. A similar pattern was observed in the DLQI scores. Mutation carriers reported significantly greater improvements in dermatology-related quality of life over time, as shown by lower DLQI scores (F (4, 280) = 3.5332, *p* = 0.00784). The difference emerged as early as month 1 and persisted throughout the observation period. Together, these findings suggest that the *CARD14 rs34367357* mutation may be associated with a more rapid and substantial improvement in patient-reported and objective measures of psoriasis severity, particularly with respect to skin surface involvement and quality of life, while PASI reduction did not reach statistical significance.

We examined treatment response stratified by biologic agent class: TNF-alpha inhibitors versus interleukin inhibitors (including IL-17, IL-12/23, and IL-23 inhibitors). Both drug classes demonstrated progressive efficacy over time, and significant differences were observed across all three clinical outcome measures (Figure 2). PASI scores declined steadily in both treatment groups; however, patients receiving IL-inhibitors showed a more pronounced reduction from month 4 onward, with the interaction term reaching statistical significance (F (4, 280) = 3.8421, *p* = 0.00480). By month 10, PASI values in the IL-inhibitor group approached minimal disease activity. Similarly, BSA clearance was more rapid in the IL-inhibitor group (F (4, 280) = 3.9854, *p* = 0.00897), with differences becoming apparent from month 4. DLQI improvement was also significantly faster in patients treated with IL inhibitors (F (4, 280) = 4.6011, *p* = 0.00200), suggesting a more rapid improvement in quality of life. The steepest drop occurred within the first month, indicating an early perceived benefit. These results support the hypothesis that interleukin inhibition may offer superior short-term skin clearance and QoL recovery compared to TNF-alpha inhibition in this patient population.

The analysis showed no significant difference in BMI between carriers of the *CARD14 rs34367357* mutation and individuals with the wild-type allele (*p* = 0.845), indicating that body weight was not a confounding factor in assessing treatment response. The similar BMI distribution in both groups allows for the exclusion of BMI ≥ 25 kg/m^2^ as a driver of the observed differences in therapeutic outcomes. To explore the influence of nutritional status on therapeutic response, patients were ranked based on simplified BMI categorization into two subgroups: normal weight with BMI < 25 kg/m^2^ and overweight or obese with BMI ≥ 25 kg/m^2^. While baseline PASI, BSA, and DLQI scores did not significantly differ between groups, notable trends emerged in initial treatment kinetics. Patients with BMI ≥ 25 demonstrated a slower clinical improvement trajectory compared to those with BMI < 25, particularly in terms of PASI scores (Figure 3). A significant interaction between time and nutritional status was observed for PASI (F (4, 280) = 2.8606, *p* = 0.02385), indicating that individuals with normal weight achieved better disease clearance over the 10-month period. While the baseline PASI values were similar across both groups, the reduction was more definite among patients with a BMI < 25, especially between months 1 and 4. For BSA and DLQI, although the trends suggested greater numerical improvement in the normal-weight subgroup, the interaction terms did not reach statistical significance (BSA: *p* = 0.52784; DLQI: *p* = 0.09348). In the DLQI assessment, quality-of-life improvements appeared consistently more substantial in the lower BMI group across all timepoints. These findings suggest that excess body weight may be associated with a modest attenuation of therapeutic efficacy, particularly in the initial stages of treatment. This supports prior evidence linking obesity to systemic inflammation, altered pharmacokinetics of biologics, and lower treatment response rates in psoriasis.

Patients who initiated biological treatment during the warmer months experienced significantly faster and more pronounced improvement in disease activity and quality of life (Figure 4). A significant interaction between the timepoint and treatment initiation season was observed for all three clinical parameters: PASI (F (4, 280) = 24.881, *p* < 0.00001), BSA (F (4, 280) = 6.9739, *p* = 0.00002), and DLQI (F (4, 280) = 2.7063, *p* = 0.03068). Patients who started treatment in the warm period had consistently lower PASI and BSA scores at each follow-up compared to those starting during the colder season. Although the baseline values were slightly higher in the warm-season group, their clinical trajectories diverged quickly, with more rapid clearance of lesions observed already by month one. This effect persisted throughout the 10-month observation period, suggesting a potential modifying influence of seasonality on treatment dynamics. DLQI improvements followed a similar pattern, with earlier and more substantial quality-of-life gains in patients initiating therapy during the warmer months. Conversely, in the cold-start group, a mild rebound of PASI and BSA scores was noted after month seven, coinciding with winter, which may indicate seasonal exacerbation or environmental attenuation of therapeutic effects. These observations support the hypothesis that environmental factors—such as ambient temperature, UV exposure, or behavioral shifts—may synergize with biologic treatment to enhance clinical outcomes in psoriasis.

While the three-way interaction between *CARD14* variant, treatment initiation season, and time was not statistically significant, the observed trends merit further exploration (Figure 5). Patients harboring the *CARD14 rs34367357* mutation who began biologic therapy during the warmer seasons exhibited a more consistent reduction in PASI and BSA scores over time compared to patients initiating treatment in the cold season. Although the interaction effects did not reach formal statistical significance (PASI: *p* = 0.1526; BSA: *p* = 0.0678; DLQI: *p* = 0.8415), the divergence in trajectories, particularly between months 4 and 10, suggests a potential seasonal modulation of therapeutic efficacy in genetically predisposed individuals. This pattern may reflect a form of environmental vulnerability in *CARD14*-mutated patients, related to cutaneous immune reactivity, barrier dysfunction, or circannual variation in cytokine activity. It is probable that those beginning treatment in warmer months—while initially responding well—experienced weakened efficacy during the subsequent colder months, potentially due to seasonal triggers such as reduced UV exposure, increased air dryness, or higher incidence of infections. This seasonal reduction was not observed to the same extent in patients with the wild-type *CARD14* allele, further reinforcing the hypothesis of a gene–environment interaction.

Although these findings require confirmation in larger cohorts, they raise the possibility that the timing of treatment initiation could be an important consideration in optimizing outcomes for patients with psoriasis carrying *CARD14* mutations. Further mechanistic studies are necessary to explore whether seasonal immunological shifts may differentially influence cytokine-targeted therapies in genetically defined subgroups.

## 5. Discussion

Our study provides novel insight into the influence of genetic, environmental, and metabolic factors on the clinical efficacy of biologic therapies in psoriasis. Specifically, we observed that seasonality of treatment initiation, nutrition status, and the presence of the *CARD14 rs34367357* variant may each modulate the trajectory of therapeutic response, particularly in terms of PASI and BSA reduction. While some of these findings reached statistical significance and others remained trends, the integrated analysis suggests that these factors may exert a synergistic or modifying effect on treatment outcomes and merit further exploration.

One of the most consistent and statistically significant findings in our cohort was the impact of the season of treatment initiation on clinical response. Patients who began biological therapy during the warm season exhibited significantly faster and more sustained reductions in PASI and BSA scores over the 10-month observation period, as well as earlier improvements in DLQI. These differences were evident as early as the first month of therapy and remained apparent through the final visit. This aligns with previous reports suggesting that environmental factors, such as UV exposure, temperature, and humidity, play an important role in modulating disease activity in psoriasis. For instance, exposure to natural sunlight is known to exert immunosuppressive effects, particularly by reducing dendritic cell activation and downregulating Th17/IL-23 signaling pathways—a central axis in psoriatic inflammation targeted by modern biologics [17,25].

Several studies have investigated the seasonal variability of psoriasis. Brito et al. [26] analyzed hospitalization patterns in a dermatology ward and found no differences related to season in patients with psoriasis. In contrast, Ferguson et al. [27] reported that 77% of patients experienced seasonal fluctuations in disease activity, with most exacerbations occurring in winter (67%), followed by summer (24%). In a survey conducted in northern Germany, Mrowietz et al. [28] identified four self-reported disease patterns: stable disease without seasonal variation (40.86%), unstable disease without seasonal variation (22.57%), increased winter flare frequency (30.6%), and increased summer flare frequency (5.97%). Similarly, a large retrospective analysis of 20,270 Chinese patients [29] showed that severe psoriasis during autumn and winter was associated with longer disease duration, hyperlipidemia, and smoking, whereas older age and occupations involving high sunlight exposure were linked to a lower likelihood of seasonal exacerbations. Jensen et al. [21] found that only 30% of psoriatic patients in northern and central Europe reported symptom improvement during summer, suggesting that factors other than season alone may influence disease course. The biological plausibility of this seasonal modulation is multifactorial. During colder months, reduced levels of UVB radiation translate into lower cutaneous vitamin D synthesis, which has immunomodulatory properties relevant to psoriasis pathogenesis [1,25]. In addition, cold and dry weather can compromise the epidermal barrier, leading to increased transepidermal water loss, xerosis, and microinflammation, which may act synergistically with psoriatic inflammation [30]. Increased incidence of upper respiratory tract infections and seasonal behavioral changes (e.g., indoor air heating, clothing friction, mood fluctuation) may also contribute to flare-ups or reduced treatment efficacy [20]. Recent studies have also suggested a role of gut and skin microbiota in psoriasis pathogenesis and treatment response [31,32,33]; however, this aspect was beyond the scope of our analysis, which focused on genetic, seasonal, and metabolic factors.

The effect of obesity on therapeutic outcomes is also noteworthy. In our analysis, patients with BMI ≥ 25 exhibited slower improvement in PASI scores compared to those with BMI < 25, with a statistically significant interaction between BMI and timepoint. While BSA and DLQI trends followed a similar direction, these did not reach statistical significance. These results are consistent with prior literature suggesting that obesity is associated with poorer treatment response to biologics, particularly TNF-α and IL-17 inhibitors [4,24,34]. Several mechanisms have been proposed to explain this association. Obesity is a state of chronic low-grade inflammation, with adipose tissue acting as an active endocrine organ that produces proinflammatory cytokines such as TNF-α, IL-6, and leptin—all implicated in the pathophysiology of psoriasis [35]. Furthermore, obesity may alter drug pharmacokinetics, particularly for weight-fixed biologics like etanercept or adalimumab, potentially resulting in subtherapeutic exposure in heavier individuals [36]. These findings have prompted calls for complementary lifestyle interventions in psoriasis management, particularly among obese individuals, to augment pharmacologic efficacy and improve long-term outcomes.

Functional studies of related *CARD14* mutations have shown that they activate NF-κB and upregulate inflammatory mediators in keratinocytes, implicating this pathway in disease pathogenesis [15,16,37,38,39]. Emerging data confirm that the *CARD14* variant *rs34367357* is associated with psoriasis in a Pakistani cohort, highlighting its potential role as a susceptibility allele [16]. This variant, located in the PDZ domain of *CARD14*, may facilitate NF-κB activation via enhanced signaling complex formation, consistent with the known mechanisms of other pathogenic *CARD14* mutations. Although the functional consequences of *rs34367357* specifically remain to be clarified, its association with psoriasis reinforces the importance of *CARD14* in skin inflammation [15,37,39].

Our findings may represent one of the first efforts to quantitatively assess the role of this polymorphism in modulating biologic treatment outcomes over time, particularly in relation to seasonality. Our sample size may have limited the ability to detect statistically robust interactions; however, the divergence observed in BSA and PASI trajectories, particularly between months 4 and 10, should be interpreted as a non-significant trend. This observation is noteworthy and warrants further investigation. Future studies with larger stratified samples and environmental exposure tracking (e.g., UV index, temperature, and air pollution) may help to confirm and expand on this preliminary observation. This study has several strengths, including the prospective collection of standardized clinical outcome measures (PASI, BSA, and DLQI) at regular intervals, genotyping for a known psoriasis susceptibility locus, and the consideration of multiple modifying factors such as seasonality and BMI.

However, certain limitations must be acknowledged. First, the sample size was relatively small, and subgroup analyses were underpowered, which reduces the robustness of some of the observed effects. Second, the definition of treatment initiation season was based solely on calendar months and did not account for individual variations in environmental exposure such as UV index, temperature, or humidity, which may have introduced misclassification bias. Third, body mass index was analyzed using simplified categorical thresholds (≥25 vs. <25 kg/m^2^), which do not capture the full complexity of adiposity, metabolic health, or body composition. Taken together, these limitations underscore that the results should be interpreted with caution and regarded as exploratory. Larger, prospective, and mechanistically oriented studies will be required to validate these findings and confirm their clinical relevance.

Taken together, our study highlights seasonality, obesity, and genetic background as potential modulators of biologic therapy outcomes in psoriasis. While the exploratory interaction analyses were not conclusive, the observed trends underscore the need for larger prospective studies integrating genetic and environmental variables.

## 6. Conclusions

Our findings support the hypothesis that seasonal timing, nutritional status, and genetic background may modulate the efficacy of biologic therapies in psoriasis. Patients starting treatment during the warm season and those with normal BMI experienced more rapid and sustained clinical improvement. For *CARD14 rs34367357* carriers, trends toward better outcomes were observed, but these were not statistically significant. The potential interaction between *CARD14* status and seasonality warrants further investigation. These results highlight the importance of considering contextual and host-related factors when initiating systemic therapy and may inform future strategies for personalized and seasonally optimized psoriasis care.

## Figures and Tables

**Figure 1 jcm-14-06458-f001:**
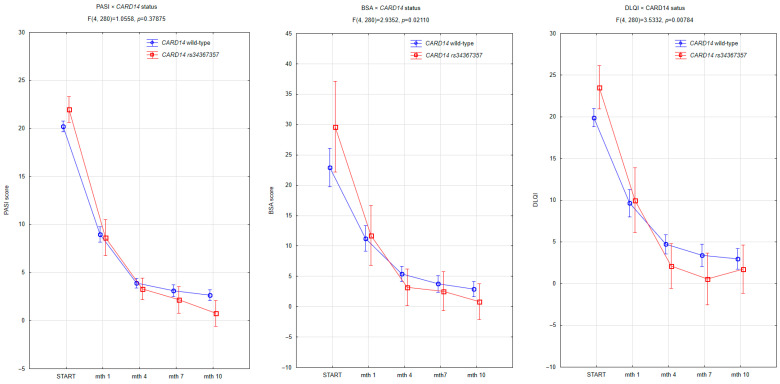
Treatment outcomes in patients with and without *CARD14 rs34367357* mutation during the first 10 months of biologic therapy. A significant interaction was observed between *CARD14* mutation status and treatment time for BSA (*p* = 0.02110) and DLQI (*p* = 0.00784), but not for PASI (*p* = 0.37875). Error bars represent 95% confidence intervals.

**Figure 2 jcm-14-06458-f002:**
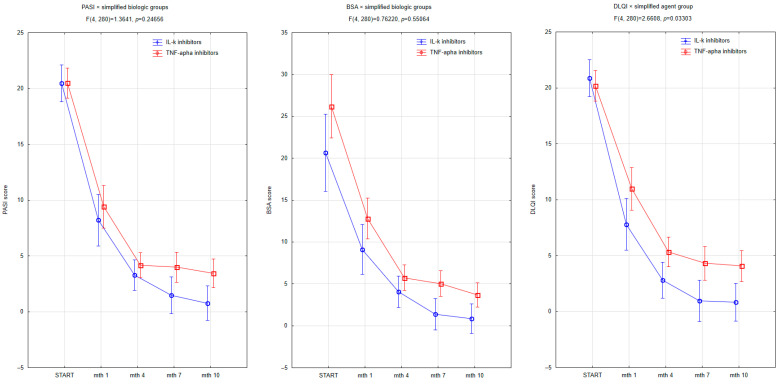
Trajectories of PASI, DLQI, and BSA improvement during biologic therapy with interleukin inhibitors versus TNF-alpha inhibitors. Patients receiving IL inhibitors demonstrated significantly greater reductions in PASI (*p* = 0.00480), DLQI (*p* = 0.00200), and BSA (*p* = 0.00897) across the 10-month follow-up. The most notable differences emerged after month 4, indicating superior intermediate-term efficacy of IL-targeted biologics. Values are presented as means with 95% confidence intervals.

**Figure 3 jcm-14-06458-f003:**
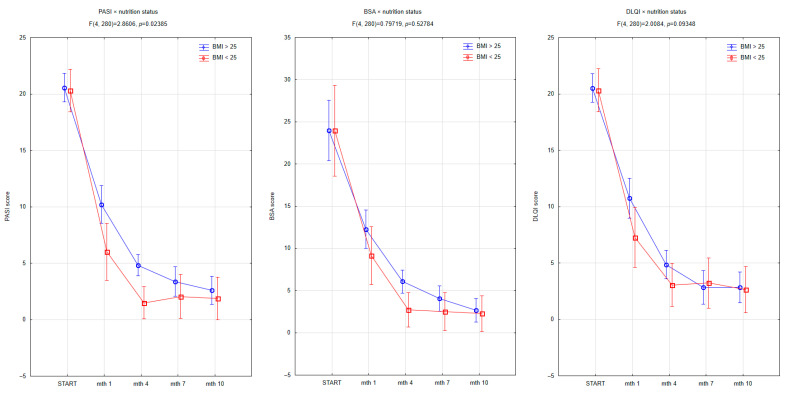
Changes in PASI, BSA, and DLQI scores over 10 months of biologic therapy according to nutritional status (BMI ≥ 25 vs. BMI < 25). Patients with BMI < 25 demonstrated significantly faster improvement in PASI scores, as evidenced by a significant time × BMI interaction (F (4, 280) = 2.8606, *p* = 0.02385). While trends toward greater improvement in BSA and DLQI were also observed in the lower BMI group, these differences did not reach statistical significance (BSA: *p* = 0.52784; DLQI: *p* = 0.09348). Error bars represent 95% confidence intervals.

**Figure 4 jcm-14-06458-f004:**
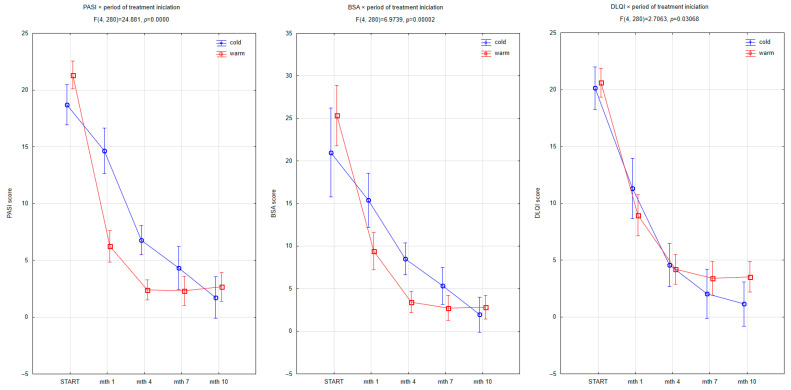
Changes in PASI, BSA, and DLQI scores over the first 10 months of biologic therapy depending on the season of treatment initiation (warm vs. cold period). Patients starting treatment during the warm season showed significantly faster and more pronounced clinical improvement in all three parameters. A significant interaction between timepoint and treatment initiation season was observed for PASI (F (4, 280) = 24.881, *p* < 0.00001), BSA (F (4, 280) = 6.9739, *p* = 0.00002), and DLQI (F (4, 280) = 2.7063, *p* = 0.03068). A plateau or mild deterioration in PASI and BSA scores was observed among patients who began treatment in the warm period and continued into the winter months.

**Figure 5 jcm-14-06458-f005:**
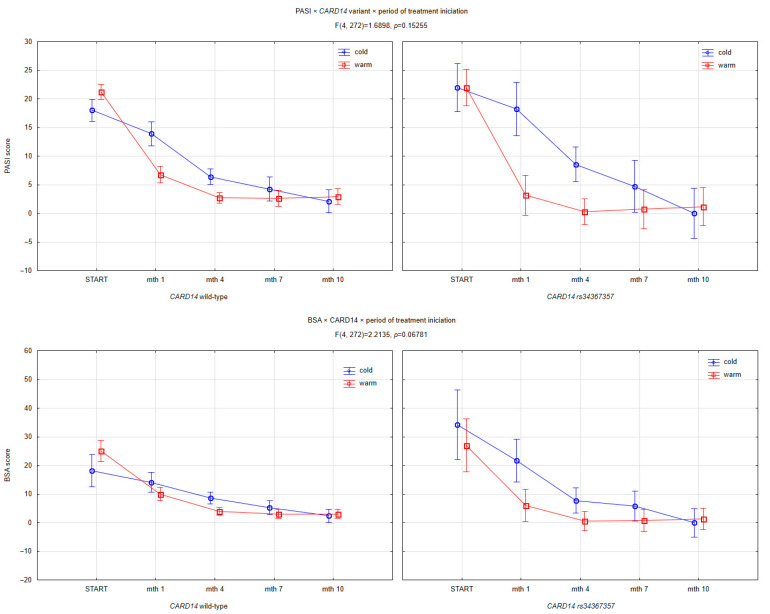
Interaction between *CARD14 rs34367357* variant, season of treatment initiation (cold vs. warm), and time on clinical outcomes over 10 months of biologic therapy. Panels show the evolution of PASI (top row), BSA (middle row), and DLQI (bottom row) scores stratified by *CARD14* status (wild-type on the left; *rs34367357* variant on the right) and by season of treatment initiation. While no statistically significant three-way interaction was observed for any parameter (PASI: F (4, 272) = 1.6898, *p* = 0.15255; BSA: F (4, 272) = 2.2135, *p* = 0.06781; DLQI: F (4, 272) = 0.35352, *p* = 0.84148), patients with the *CARD14 rs34367357* variant initiating treatment during the warm season demonstrated a trend toward more sustained improvement in BSA and PASI scores over time, particularly between months 4 and 10. This suggests a potential seasonal modulation of therapeutic efficacy in genetically predisposed individuals.

**Table 1 jcm-14-06458-t001:** Baseline characteristics, biologic treatment profiles, and clinical parameters among patients with and without the *CARD14 rs34367357* mutation. Mutation carriers demonstrated significantly earlier onset of psoriasis and better baseline quality of life (DLQI). No significant differences were observed in baseline PASI, BSA, BMI, sex distribution, or age at treatment initiation. Biologic agents were similarly distributed between groups. Agents were also categorized into simplified classes, with TNF-alpha inhibitors being most frequently used. Values are presented as mean ± standard deviation or patient counts. Statistical comparisons were conducted using *t*-tests or chi-square tests, where applicable.

Variable	No Mutation (Mean ± SD or *n*)	Mutation (Mean ± SD or *n*)	*p*-Value
females	22	6	0.4115
males	39	5	0.4115
BMI	27.1 ± 6.3	26.7 ± 4.4	0.8051
age of psoriasis onset (years)	22.4 ± 14.9	13.9 ± 8.4	0.0134
age at treatment start (years)	35.4 ± 14.7	36.7 ± 14.9	0.7887
baseline PASI	20.2 ± 4.4	22.0 ± 4.7	0.2664
baseline BSA	22.9 ± 10.9	29.6 ± 19.3	0.2858
baseline DLQI	19.9 ± 4.3	23.5 ± 4.5	0.0265
Biologic agents			
TNF-alpha inhibitors	36	7	
IL-23 inhibitors	6	0	
IL-12/23 inhibitors	6	0	
IL-17 inhibitors	13	4	

Abbreviations: BMI—Body Mass Index; PASI—Psoriasis Area and Severity Index; BSA—Body Surface Area; DLQI—Dermatology Life Quality Index; TNF—Tumor Necrosis Factor; IL—interleukin; SD—Standard Deviation.

## Data Availability

The data presented in this study are available on request from the corresponding author due to privacy and ethical restrictions related to patient confidentiality.

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
