# Peer review of "Impact of CARD14 rs34367357 Mutation, Nutrition Status, and Seasonality on the Response to Biologic Therapy in Psoriasis—A Retrospective Observational Single-Center Study"

_jcm, 2025, doi:10.3390/jcm14186458_

Round 1
Reviewer 1 Report
Comments and Suggestions for Authors
As a main view, the manuscript is well written and very up to date, bringing new factors to this important topic.
Psoriasis is a current and important topic, as it leads not only to dermatological problems but also directly impacts patients' self-esteem. However, as the manuscript rightly points out, this is not a simple condition. It is multifactorial and complex, where genetic factors, lifestyle, psychological factors and medical conditions combine, making its treatment difficult. The authors highlighted these factors very well, but they failed to address a factor that has become increasingly important today: the human microbiome. In this sense, although this is not the focus of the work, the inclusion of a few paragraphs and articles, as well as a brief discussion, could enrich the article.
Furthermore, as a suggestion, the authors should address the fact that during the cold season (as they describe autumn and winter), people tend to take more hot baths, which significantly removes the skin's hydrolipidic mantle, leading to an imbalance in both barrier function and cutaneous dysbiosis. This can be corroborated by the results regarding the presence of IL-17, directly linked to cutaneous dysbiosis, and explain the fact that when the treatment starts at warm season the results are better.
Finally, regarding the nutritional factors, obesity is also related to intestinal factors, mainly nutrients absorption and microorganism alterations, so to point this at the discussion, could foment a better understanding of psoriasis as a multidisciplinary alterations that need to be treat by broad view , highlighting the importance of this manuscript.
The graphics should be larger to make them easier to see.
Author Response
Comment:
As a main view, the manuscript is well written and very up to date, bringing new factors to this important topic.
Psoriasis is a current and important topic, as it leads not only to dermatological problems but also directly impacts patients' self-esteem. However, as the manuscript rightly points out, this is not a simple condition. It is multifactorial and complex, where genetic factors, lifestyle, psychological factors and medical conditions combine, making its treatment difficult. The authors highlighted these factors very well, but they failed to address a factor that has become increasingly important today: the human microbiome. In this sense, although this is not the focus of the work, the inclusion of a few paragraphs and articles, as well as a brief discussion, could enrich the article.
Furthermore, as a suggestion, the authors should address the fact that during the cold season (as they describe autumn and winter), people tend to take more hot baths, which significantly removes the skin's hydrolipidic mantle, leading to an imbalance in both barrier function and cutaneous dysbiosis. This can be corroborated by the results regarding the presence of IL-17, directly linked to cutaneous dysbiosis, and explain the fact that when the treatment starts at warm season the results are better."
Finally, regarding the nutritional factors, obesity is also related to intestinal factors, mainly nutrients absorption and microorganism alterations, so to point this at the discussion, could foment a better understanding of psoriasis as a multidisciplinary alterations that need to be treat by broad view , highlighting the importance of this manuscript.
Responce:
We sincerely thank the Reviewer for these thoughtful suggestions. We thank the reviewer for highlighting the potential role of the microbiome. We have added a short note in the Discussion acknowledging that gut and skin microbiota may also contribute to psoriasis pathogenesis and treatment outcomes, although this was beyond the scope of the present study (page 11, lines 366-368).
Comment:
The graphics should be larger to make them easier to see.
Responce:
All figures have been resized and reformatted to improve readability and presentation.
We believe these additions have enriched the manuscript and broadened the multidisciplinary perspective, in line with the Reviewer’s valuable recommendations.
Reviewer 2 Report
Comments and Suggestions for Authors
The manuscript is scientifically sound and addresses a topic of interest, integrating genetics, seasonality, and nutritional status in the biological treatment of psoriasis. However, some areas require improvement:
- The text is well-written, but very long sentences reduce readability.
- The rationale for studying the CARD14 rs34367357 variant should be better explained. Furthermore, reference should be made to the possible role of hormonal and sex differences in psoriasis, a factor known to influence disease response and severity. (See, for example, 10.3390/jcm14020582)
- More details are needed regarding the handling of any missing data, sequencing quality control, and potential dose adjustments for biologics.
- The data are well-presented, but Table 1 is difficult to follow. It is recommended that it be reorganized to clearly distinguish between the mutant and wild-type groups without redundancies in drug categories.
- The conclusions are consistent, but some results (e.g., "faster response in mutation carriers") should be described as trends that are not statistically confirmed. It is useful to strengthen the clinical part by suggesting practical implications: for example, considering the time of year when starting therapy, or interventions on body weight to improve response.
- Limitations: These should be emphasized more forcefully: small sample size, underpowered subgroup analyses, approximate definition of "season" based only on the calendar, overly simplified BMI classification.
These improvements could refine the work and make it more scientifically sound.
Author Response
We sincerely thank Reviewer for the constructive and detailed feedback, which helped us to improve the clarity, scientific rigor, and clinical relevance of our manuscript. In response to the comments:
Comment:
- The text is well-written, but very long sentences reduce readability.
Response:
To improve readability, we carefully revised the manuscript and shortened several long sentences.
Comment:
- The rationale for studying the CARD14 rs34367357 variant should be better explained. Furthermore, reference should be made to the possible role of hormonal and sex differences in psoriasis, a factor known to influence disease response and severity. (See, for example, 10.3390/jcm14020582)
Response:
In the revised version, we expanded the Introduction to provide a clearer rationale for analysing the CARD14 rs34367357 variant, highlighting its reported association with psoriasis susceptibility, early disease onset, and familial forms of the disease (page: 2, lines: 72-79).
We incorporated a discussion of hormonal and sex-related influences on psoriasis severity and biologic response, with reference to the suggested article (J Clin Med. 2025;14(2):582). These revisions improve the contextual background and scientific justification of our study (page: 2, lines: 63-71).
Regarding sex-related differences, we performed an additional analysis of treatment outcomes by sex. Repeated-measures ANOVA showed no significant interaction between sex and time for PASI (p = 0.1825), BSA (p = 0.9138), or DLQI (p = 0.9337). Both male and female patients demonstrated comparable improvement trajectories over the 10-month observation period (page: 5, lines: 182–186).
Comment:
- More details are needed regarding the handling of any missing data, sequencing quality control, and potential dose adjustments for biologics.
Response:
We have revised the Methods section to clarify these points. Specifically, we now state that a complete-case analysis was performed, as only patients with full PASI, BSA, and DLQI data across all scheduled timepoints (0, 1, 4, 7, and 10 months) were included. Because patients in the NHF B.47 program are required to attend scheduled visits and undergo standardized assessments, missingness was negligible and no imputation or casewise/listwise deletion was necessary.
Regarding sequencing, we have specified that CARD14 genotyping was performed by Sanger sequencing . In accordance with the laboratory’s standard protocol, each sample was tested in triplicate to ensure reproducibility.
Finally, we clarified that all biologic agents were administered strictly according to their respective Summaries of Product Characteristics (SMPC), within the NHF B.47 program framework, and no off-label dose adjustments were performed during the study period.
These revisions are now included in the Methods section (page: 3, lines: 128-129 and 139-141; page: 4, lines: 148-151.
Comment:
- The data are well-presented, but Table 1 is difficult to follow. It is recommended that it be reorganized to clearly distinguish between the mutant and wild-type groups without redundancies in drug categories.
Response:
Table 1 has been reorganized to improve clarity. Demographic and baseline clinical characteristics are now presented in a concise format, with separate columns for patients with and without the CARD14 rs34367357 mutation. Biologic agents have been grouped into main therapeutic classes (TNF-α, IL-23, IL-12/23, and IL-17 inhibitors) to avoid redundancies (pages: 4 and 5).
Comment:
- The conclusions are consistent, but some results (e.g., "faster response in mutation carriers") should be described as trends that are not statistically confirmed. It is useful to strengthen the clinical part by suggesting practical implications: for example, considering the time of year when starting therapy, or interventions on body weight to improve response.
Response:
We thank the Reviewer for this important observation. In the revised version, we clarified that the differences in PASI and DLQI trajectories among CARD14 rs34367357 carriers represent non-significant trends rather than statistically confirmed findings. This clarification was added to the Discussion (page 11, lines 366-368) and Conclusions (page: 12, lines: 422-426) to ensure accurate interpretation of our results.
In addition, we expanded the clinical perspective of the manuscript by highlighting practical implications. Specifically, we now emphasize that the timing of biologic therapy initiation (warm vs. cold season) and weight management interventions may be relevant considerations in optimizing treatment outcomes in patients with psoriasis.
Comment:
- Limitations: These should be emphasized more forcefully: small sample size, underpowered subgroup analyses, approximate definition of "season" based only on the calendar, overly simplified BMI classification.
Response:
The Limitations section has been revised and expanded to more clearly emphasize the constraints of our study, including the small sample size, the limited statistical power of subgroup analyses, the approximate definition of “season” based on calendar months rather than environmental exposure measures, and the simplified dichotomization of BMI (page: 12, lines: 422-426). These revisions highlight that our findings should be regarded as exploratory and hypothesis-generating.
We believe these revisions have addressed all concerns raised by the Reviewer and have substantially improved the quality of the manuscript.